# TREE SEARCH FOR SIMULTANEOUS MOVE GAMES VIA EQUILIBRIUM APPROXIMATION

## ABSTRACT

Neural network supported tree-search has shown strong results in a variety of perfect information multi-agent tasks. However, the performance of these methods on partial information games has generally been below competing approaches. Here we study the class of simultaneous-move games, which are a subclass of partial information games which are most similar to perfect information games: both agents know the game state with the exception of the opponent's move, which is revealed only after each agent makes its own move. Simultaneous move games include popular benchmarks such as Google Research Football and Starcraft.

In this study we answer the question: can we take tree search algorithms trained through self-play from perfect information settings and adapt them to simultaneous move games without significant loss of performance? We answer this question by deriving a practical method that attempts to approximate a coarse correlated equilibrium as a subroutine within a tree search. Our algorithm works on cooperative, competitive, and mixed tasks. Our results are better than the current best MARL algorithms on a wide range of accepted baselines.

## 1 INTRODUCTION

Multi-agent reinforcement learning (MARL) algorithms train multiple agents interacting in a shared environment. The challenge compared to single-player reinforcement learning is that, for each agent, the actions of the other agents are influencing the received rewards, and therefore the evolution of other agents' policies during training causes the environment to appear non-stationary from the perspective of a single agent. While multi-agent reinforcement learning methods have achieved great successes for tasks where agents receive full information (see e.g., the survey Zhang et al. (2021)), successes in partially observable settings have been more muted.

Our study focuses on a subset of partial information tasks most similar to perfect information tasks, namely simultaneous move tasks; the only information withheld in simultaneous move tasks is the actions of the other players. Contemporary simultaneous-move algorithms are largely designed for either cooperative Yu et al. (2021) or competitive tasks Schrittwieser et al. (2019). Most contemporary algorithms take advantage of assumptions present when all agents are working together (cooperative tasks) or agents are competing against one another (competitive tasks) in their design. The algorithm we develop and present in this study is one of a small group Lowe et al. (2017) that can be applied to both competitive and cooperative tasks.

Our new MARL algorithm combines a popular method for competitive tasks, namely deep-Monte Carlo Tree Search (d-MCTS), with online no-regret learning to approximate a coarse correlated equilibrium (CCE), a concept from game-theory. As we explain later, playing according to a CCE gives you performance guarantees against any opponent in competitive tasks. Therefore even though our method is trained purely through self-play, we demonstrate strong performance against all contemporary algorithms in the competitive setting, even against algorithms trained with human-injected knowledge such hand-coded opponents.

Specifically, we demonstrate that our method surpasses several leading multi-agent reinforcement learning (MARL) algorithms—including Policy Space Response Oracles (PSRO), Multi-Agent Proximal Policy Optimization (MAPPO), and Multi-Agent Deep Deterministic Policy Gradient (MADDPG)—as well as numerous other competing approaches. Notably, MAPPO and MADDPG

have previously established strong performance benchmarks among competitor algorithms Yu et al. (2021); Lowe et al. (2017). Our agents learn superior policies in both cooperative and competitive settings, achieving win rates exceeding 80% in head-to-head evaluations against these baselines. We evaluated our algorithm on 17 simultaneous-move games with publicly available code that have been studied in prior research. Our method outperforms all other tested algorithms on 15/17 of these benchmarks and is competitive on the remaining two (though it is trained with self-play and the competitor algorithms use human knowledge injection).

## 2 BACKGROUND

We define an $N$-player stochastic game (SG) as $(S, H, \{A_i\}_{i \in N}, T, \{U_i\}_{i \in N}, \gamma)$, where $\mathbf{S}$ is the set of all states shared by all $\mathbf{N}$ players, $\mathbf{H}$ is the horizon (the maximum number of time steps), $\mathbf{A_i}$ is the action space for player $i$ yielding the decomposition $\mathbf{A} := \mathbf{A_1} \times \cdots \times \mathbf{A_N}$, $\mathbf{T} : (S \times A) \to S'$ is the state transition function, $\mathbf{U_i} : (S \times A) \to \mathcal{R}$ is the utility function for each player $i \in N$, and $\gamma$ is the discount factor. Finally, we denote by $\Delta(S)$ a distribution over the starting states.

### 2.1 BACKGROUND: DEEP MARL TRAINING

Our goal is to train a set of agents, defined by their policies, $\{\pi_i\}_{i \in N}$, where each policy, $\pi_i : S \to A$. Here the simultaneous-move nature of the game will come through, as each policy $\pi$ takes only the state (not the actions of the other players that are unknown) as arguments. The game then evolves from state $s$ with the joint action $(\pi_1(s), \ldots, \pi_n(s))$ played.

Training through deep reinforcement learning is usually comprised of two iterating steps: data generation and network training. Data generation aims to create a data set, $\mathcal{D}_t$, that the NN samples from to train. A NN is used to approximate the value function and policy function, given the following loss functions

$$L(\theta_t) = \frac{1}{|\mathcal{D}_t|} E_{s \sim \mathcal{D}_t} L(g_t(s), \hat{g_{\theta_t}}(s))$$

where $g_t$ is the value or policy function at time step $t$, $\hat{g_{\theta_t}}$ is the value or policy prediction of the network at time step $t$ , $L$ is an appropriate loss function, and $\theta_t$ represents the network parameters at time step $t$. The data generation is usually accomplished through repeated interaction with the stochastic game. The recorded interactions are then used to re-train the same NNs Lee et al. (2022).

### 2.2 NO-REGRET LEARNING AND ONLINE LEARNING

There is a close relationship between multi-agent learning and game theory which we will exploit.

The concept of an equilibrium provides a strong learning objective in multi-agent settings. The most popular equilibrium, the Nash Equilibrium, describes a set of strategies in a two player zero-sum (2p0s) game in which neither player gains any benefit from changing strategies. While computing a NE is ideal, it was shown to be PPAD-complete even in 2p0s games Nisan et al. (2007).

Less restrictive forms of equilibrium can be approximated using no-regret learning. In this study, we will attempt to approximate an $\epsilon$-coarse correlated equilibrium (CCE). A CCE is defined as

$$\forall i, a_i' \quad E_{s \sim \sigma} \, c_i(a) \leq E_{s \sim \sigma} \, c_i(a_i', a_{-i}) + \epsilon$$

where $i$ represents a player, $a_i'$ represents an action different from the recommended action, $a$, and $c_i$ represents the cost of following a strategy. At first glance, this might look the normal definitions of mixed strategy Nash equilibrium, but observe that the joint state $s$ is being sampled from some distribution $\sigma$. Thus this definition says that there exists a joint distribution of the strategies such that deviations do not benefit each player, provided that all the remaining players sample from the same correlated distribution.

It is known that if all players in a SM game use no-regret online learning, then their time-averaged policies converge to the set of CCEs Tardos (2020); Roughbarden (2016). Here we define the regret at time step $T$ is defined as

$$R_T = \max_{i \in [K]} \mathbb{E} \left[ \sum_{t=1}^{T} l_{t, I_t} - \sum_{t=1}^{T} l_{t, i} \right]$$

where the player has an action space of size $K$, and $l_{t,k}$ represents the loss experienced at time step $t$ for action $k \in K$. It may seem puzzling that individual actions by each agent using a no-regret learner made independently converge to a correlated action distribution, but note that agents are effectively responding to each other so that correlation can be introduced into their time-averaged policies.

No-regret learning measures the difference in loss compared to the best single action in hindsight. Successful learning in this framework provides guarantees that the regret grows sub-linearly with respect to $T$ in expectation or with high probability. Below, we will utilize no-regret learning algorithms, EXP-IX Neu (2015) and EXP-WIX Kocák et al. (2016) which are known to have the property of no-regret learning with high probability.

### 2.3 WHY COARSE CORRELATED EQULIBRIUM?

Standard MCTS is tailored to finding min-max solutions. This approach works well for zero-sum, perfect information games like chess or Go, but when we move into the realm of partial information games, the min-max paradigm becomes inappropriate. In these games, players don't have complete information about the game state or their opponents' actions, and the optimal strategy often involves probabilistic decision-making to account for this uncertainty. Moreover, in multi-player or general-sum games, the strict adversarial assumption of min-max doesn't hold. Therefore, to effectively use MCTS for partial information games, we need to modify the algorithm to converge to a different solution concept, one that is more appropriate for the game-theoretic nature of these scenarios.

The computational limitations of Nash Equilibrium Nisan et al. (2007) suggest Correlated Equilibrium (CE) and Coarse Correlated Equilibrium (CCE).

Among these, CCE stands out as a particularly natural choice: the key advantage of CCE lies in its compatibility with no-regret learning dynamics where players only need to observe their own payoffs, not the entire game structure or other players' actions.

## 3 RELATED WORK

All MARL algorithms fall between two extremes: decentralized methods and centralized methods. Decentralized methods train agents simultaneously but independently. Independent Q-learning (IQL) Tampuu et al. (2017) and independent proximal policy optimization (IPPO) de Witt et al. (2020) are primary examples of decentralized algorithms. Each agent repeatedly improves their Q-value approximations, in the case of IQL, or their policy, in the case of IPPO, through repeated interactions with the environment.

On the other hand, centralized methods learn a policy over the joint action space, but are largely restricted to environments where agents share reward functions. A middle ground between the two extremes are algorithms with centralized learning and decentralized execution (CTDE). One instance is multi-agent proximal policy optimization Yu et al. (2021) (MAPPO). MA-PPO demonstrated superior performance to other popular CTDE MARL algorithms such as Simplified Action Decoder (SAD), Value Decomposition (VDN) and QMIX Rashid et al. (2018) plus its variants on several benchmarks. These algorithms are restricted to cooperative environments.

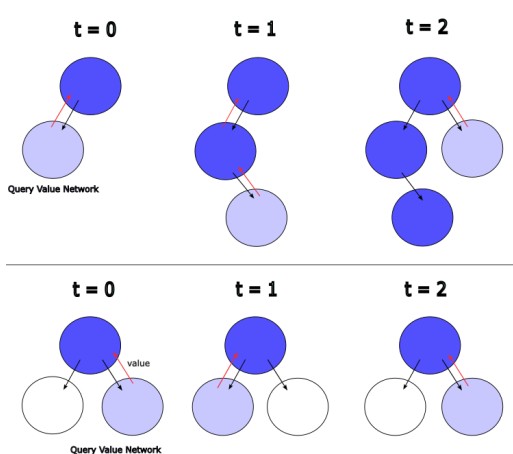

Figure 1: (Top) provides a visualization of how deep MCTS estimates the value and policy of a given node. (Bottom) provides a visualization for how our method (NN-CCE) estimates the value and policy of a given node. Each node is a state. A forward connected black edge from one node to another indicates an action connects the two states. Red backwards edges indicates that a value estimate is passed from one node to another

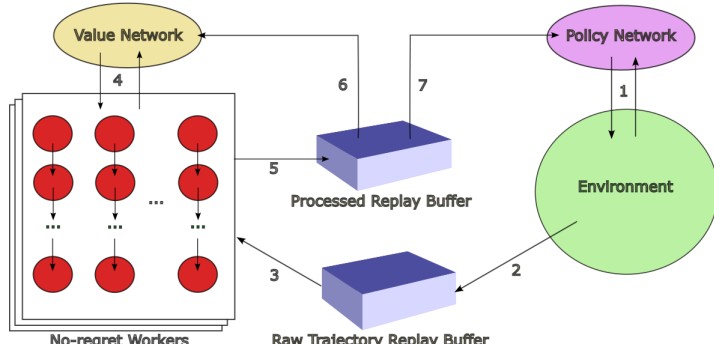

Figure 2: An overview of the working parts of our methodology and their interactions. A black directional arrow indicates that information is sent unilaterally from one entity to the other. Each number corresponds to a black arrow, and are referenced during our explanation in the methodology section.

Several studies have attempted an obvious expansion from purely cooperative to cooperative-competitive environments: behave selfishly. Hoever, it turns out that agents that behave selfishly (i.e. do not consider the policies or actions of other players) will encounter identical states with different value estimates and will have difficulty learning in mixed cooperative and competitive environmentsTampuu et al. (2017); Zawadzki et al. (2014); Lee et al. (2022)).

Another method for branching into competitive environments is to freeze the policies of certain agents during training. This way, the environment becomes stationary with respect to a single agent Vinyals et al. (2019). A common implementation of this concept is neural fictitious self-play Heinrich & Silver (2016), where an agent plays against frozen past iterations of themselves and the pool of past policies grows during training.

Next, there is a variation of policy freezing where agents either have explicit access to, or maintain their own approximation of, other agent policies. A popular example of such an algorithm is Multi Agent Deep Deterministic Policy Gradient (MADDPG)Lowe et al. (2017). This directly addresses the problem of non-stationary and allows for training on cooperative, competitive, and mixed environments. We focus our performance comparisons against MADDPG as it applies to many of the problem to which our proposed method applies.

Another approach some studies take is to attempt to approximate an equilibrium. Counterfactual regret minimization Zinkevich et al. (2008); Neller & Lanctot (2013) provides a powerful algorithm for approximating a Nash-equilibrium in 2p0s tasks. The algorithm aims to minimize regret. Zinkevich et al. (2008); Neller & Lanctot (2013) demonstrate that by attempting to minimize the regret, they also minimize the exploitability of their policy thus approximating a Nash-equilibrium. Policy Space Response Oracles (PSRO) Lanctot et al. (2017) addresses poor convergence in multi-agent settings due to other agents' policies. At each iteration, a new policy is added that approximates the best response so the meta-strategy of the other players. It has several variations such as joint PSRO (jPSRO) that are improvements upon the base algorithm. The limitation for equilibrium approximation algorithms is that they are not easily applied to tasks with larger state and action spaces. The majority of testing regarding such algorithms have been on small tasks.

Finally there is the method that is most akin to ours, deep Monte Carlo Tree Search (d-MCTS) and its variants Schrittwieser et al. (2019), Silver et al. (2017). D-MCTS utilizes neural network guided simulations at every encountered state to estimate a policy and value, which in turn become training data for future iterations of the neural networks. We provide explicit comparisons for how our methodology differs from d-MCTS methods.

## 4 METHODOLOGY

Our method can be summarized in Figure 2. It is comprised of four main pieces (value network, policy network, environment, and no-regret workers) and two replay buffers. All entities and replay buffers act asynchronously of one another and remain idle if they do not have an ongoing job.

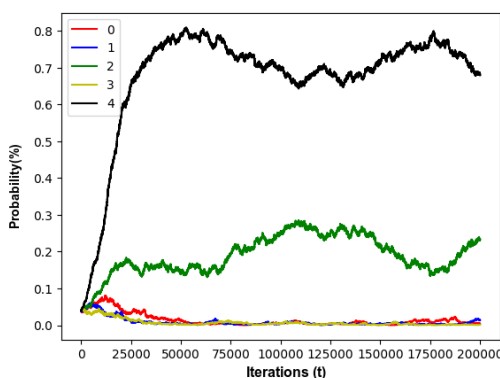 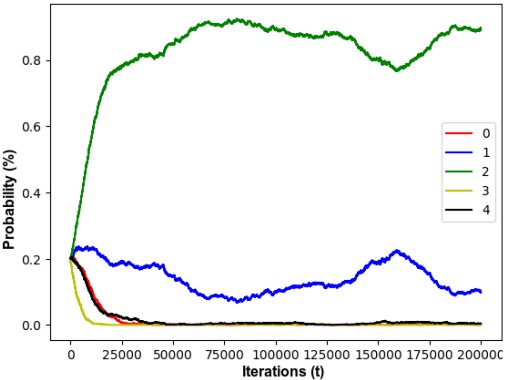

Figure 3: The probabilities of actions as a function of time for a single state in from the Multi-Particle Environment (MPE). In this state there are two players, one with an action space of 125, the second with an action space of 5. (Left) Probability change of actions for player one with an action space of 125, only 5 of the 125 actions are shown to reduce clutter in the graph. (Right) Probability changes of actions for player two displaying 5 of the 5 total actions.

In Figure 2 we see numbers noting the relationship between each of these entities if a relationship exists. We refer to these numbers in the description below. In (1), we begin with a standard inter-action between policy network and environment, where the environment sends the policy network a state, $s$, and the policy network returns the action to take, $a = \pi(s)$. Once a full trajectory has been collected this way from the environment+policy interaction, the trajectory is passed (2) to the Raw Trajectory Replay Buffer. Here, all unprocessed trajectories are stored until they are passed (3) to an available no-regret worker. Each no-regret worker takes $K$ trajectories and interacts (4) with the value network to use no-regret learning in order to estimate a value and policy for each state in each trajectory. As the worker finishes processing a trajectory, the processed trajectory is sent (5) to a separate replay buffer, the Processed Replay Buffer. Finally, the value and policy network will periodically sample data points from the processed replay buffer (6,7) in order to update their own value and policy estimates, respectively.

Many of the steps outlined above are identical to the process used in other asynchronous reinforcement learning algorithms, including deep-MCTS. The novelty of our algorithm originates from the introduction of no-regret workers into the loop and these workers' interaction with the value network. We discuss these in more depth in the next subsection.

## 4.1 NO-REGRET WORKERS AND THE VALUE NETWORK

We discuss the connection (4) here. Standard MCTS relies on repeated interaction to estimate the value and policy of a given state, $s$. Each interaction entails simulating future states, estimating the value of the future states using a value network, and updating the value and policy of $s$ using the estimated value. Let us refer to the complete interaction of estimating the a value and policy for a given state/node as "processing the state/node."

Our algorithm utilizes multiple EXP3-IX Neu (2015) instances. Let us define $K$ as the number of actions for each player and $T$ as the total number of time-steps. The core of the EXP3-IX algorithm revolves around repeated interaction between players and requires each player tracking accumulated losses for each players individually. For a set of EXP3-IX instances, $E_s$, at a given state, $s$, all instances begin with a cumulative loss vector,

$$\hat{L}_{t=0,i} = \overrightarrow{0}, \quad \forall i \in [N], \quad |\hat{L}_{t,i}| = K$$

At time-step $t \in [T]$ all players will sample will sample an action, $a_i \ \forall i \in [N]$ using their loss vectors using equation 2. We define the joint action of all players as $A = \{a_1 \dots a_N\}$. The joint action is passed to the value network and the value network outputs a vector of values, $\tilde{V}_t = \{\tilde{v}_{t,1}, \dots, \tilde{v}_{t,N}\}$.

We assume that $v_{t,i} \in [0,1] \ \forall i \in [N]$. This allows us to easily compute a loss vector $\hat{L}_t = \{L_{t,i}|L_{t,i} = 1 - v_i \ \forall i \in [N]\}$. We then utilize each players' current loss to update the cumulative loss vector via Equation 4

The value and policy estimates of each player, $i$, at time step $t$ are updated via equation 5 and equations 4 and 2 respectively. We use the values for the hyper-parameters $\eta$ and $\gamma$ as explicitly defined in Neu (2015). As an intuition: higher incurred loss for an action results in the action being selected less often in future iterations.

At the end of $T$ time-steps a policy estimate and value estimate are created by time averaging the policy and value at each time step (equations 3 and 5).

Our method deviates from standard MCTS in two major ways. First, we do not use UCB-score Schrittwieser et al. (2019) nor do we use visit count to determine a value and policy estimate. Second, in order to evaluate the value of a given node, our method relies much more on the value estimation provided by the value network compared to MCTS. We can see in Figure 1 that MCTS (top half) will

$$\hat{L}_{t,i} = \hat{L}_{t-1,i} + \frac{L_{t,i}}{P_{t-1,i} + \gamma} \quad (1)$$

$$P_{t,i} = \frac{\exp(-\eta \hat{L}_{t-1,i})}{\sum_{j=1}^{K} \exp(-\eta \hat{L}_{t-1,i})} \quad (2)$$

$$\hat{P}_i = \frac{1}{T} \sum_{t=1}^{T} P_{t,i} \quad \forall i \in [K] \quad (3)$$

$$\hat{L}_{t,i} = \hat{L}_{t-1,i} + \frac{L_{t,i}}{P_{t-1,i} + \gamma} \quad (4)$$

$$\hat{V}_i = \frac{1}{T} \sum_{t=1}^{T} \tilde{V}_{t,i} \quad (5)$$

visit new nodes beyond its immediate children (depth of 1), whereas we will only visit the immediate children of a given node. Each simulation is marked by a new time step ($t = 0, t = 1, \ldots$).

This is a trade-off where our method does not enjoy the benefits of experiencing rewards further down the tree, but we gain speed and parallelization benefits that cannot be achieved with multi-layer deep simulation. In other words, MCTS methods will encounter rewards during their path through the environment graph, where as our method is constrained to the rewards of the immediate child nodes. Both methods will query the value network for an estimate value of particular nodes, highlighted in light blue.

Before we discuss the benefits we gain through limiting simulation in this way, we first justify why this trade-off is necessary. Contrasted to deep MCTS, which typically uses up to 800 time steps per node Schrittwieser et al. (2019), no-regret learning requires significantly more.

Using the regret bound provided by Neu (2015), it becomes clear that 800 time steps is insufficient for this method. At 800 time steps, the theoretically guarantees provided by EXP3-IX are very poor: we have not yet found the best action.

Let us consider an example state that has an action space of 125. According to the regret bounds from Neu (2015), we would need to iterate at least 20,000 times to reliably determine a best action for each player (and there for CCE). A direct example of this can be in Figure 3. Here, we can see that the action probabilities for both players at the same state, sampled from MPE, stabilizes past 25,000 iterations.

For this reason, we truncate the immediate simulation depth to 1. Doing this allows us to greatly speed up the value estimation process. Firstly, we remove the need to access the environment, as we solely rely on value network evaluation feed back and do not use rewards obtained during simulation. This is an improvement in terms of speed over model-free deep MCTS, but not model-based deep MCTS.

Second, we parallelized the learning process across time. All deep MCTS methods use information from their simulations to choose their next action and begin simulating on the resulting state. Our method instead relies on a policy network to quickly traverse through a trajectory (shown in Figure 2), then processes all nodes simultaneously. As shown in Figure 4, both our method and MCTS are able to gather and process nodes (states) using multiple workers. Our method, however, processes all nodes in a single worker simultaneously, speeding up learning greatly.

## 4.2 UPDATING POLICY AND VALUE NETWORK

We now discuss the connections in (6) and (7). First, the policy network learns a mapping for each player, $P_i : s \rightarrow \hat{P}_i$ where $\hat{P}_i$ is estimated by Eq. (3). Second, the value network learns a mapping

for each player, $V_i : s \times a_i \to R(s, a_i) + \hat{V}_i$, where $R(s, A)$ is the reward encountered by the agent when collecting trajectories after taking action joint action $A = \{a_j \,|\forall j \in [N]\}$ at state, $s$ and $\hat{V}_i$ is the value estimate from equation 5.

### 4.3 METHODOLOGY: HIGH-LEVEL SUMMARY

We propose an effective way to combine no-regret learning with neural-network based tree search. Previous attempts to do this attempt to compute the value of each child state before learning the value of the parent Daskalakis et al. (2022). Given that, as we discussed above, usage of effective no-regret learning requires a large number of samples, this sequential approach makes training extremely time consuming.

By contrast, we propose a new sampling process in the tree search paradigm which first quickly traverses the tree using a policy neural network and then processes all states in parallel across time – without learning the values of the children before the parents. Our method greatly increases processing speed compared to the standard sequential approach.

Usage of no-regret learners allows us to build subroutines that converge to equilibria within the tree search. Standard MCTS is implicitly tailored towards a min-max solution, which makes sense in full but not partial information games. Instead, we use dynamics which converge to a coarse correlated equilibrium among players. Our method does not use any human knowledge injection (e.g., in the form of a hand-trained opponent to train against) and is trained purely through self-play.

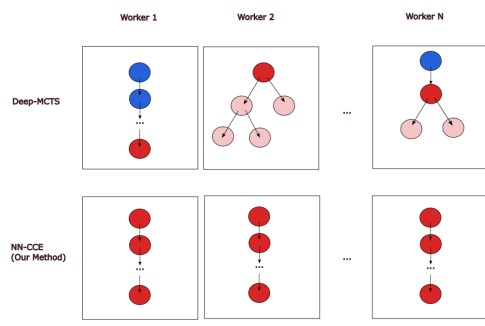

Figure 4: A diagram highlighting the differences in parallel processing between our method and standard deep-MCTS methods. Blue circles indicate nodes (states) that have had their simulations completed. Red circles indicate nodes that are currently undergoing simulation. Deep-MCTS requires each worker to serialize (rather than in parallel) the processing of each node in a trajectory. However our method first constructs the trajectories very quickly, and then processes them all at once.

## 5 POINTS OF COMPARISON

### 5.1 ENVIRONMENTS

In this study we focus on 4 main environments, where each environment contains between 2-6 unique scenarios. We define a scenario as a unique SG within an environment. Each environment was chosen because it is an open-source widely used MARL library with optimized performance to allow for fast training and was used by at least three other popular algorithms. Of all environments that fit this description, these four were the most well cited and used.

**OpenSpiel** Lanctot et al. (2019). A collection of $n$-player imperfect information games. Scenarios from this environment were small enough such

| Scen. | **NN-CCE** | MA-PPO | S-MCTS |
|---|---|---|---|
| 3v.1 | **89.00**$_{(1.50)}$ | 88.03$_{(1.06)}$ | 65.01$_{(2.21)}$ |
| CA(easy) | **90.03**$_{(1.76)}$ | 87.76$_{(1.34)}$ | 80.02$_{(2.03)}$ |
| CA(hard) | **79.03**$_{(5.85)}$ | 77.38$_{(4.81)}$ | 55.15$_{(1.22)}$ |
| Corner | **70.03**$_{(1.03)}$ | 65.53$_{(2.19)}$ | 44.19$_{(1.77)}$ |
| PS | 94.2$_{(1.06)}$ | 94.92$_{(0.68)}$ | 78.09$_{(1.23)}$ |
| RPS | 75.8$_{(1.99)}$ | 76.83$_{(1.81)}$ | 65.55$_{(0.50)}$ |

Table 1: Success rate comparison between NN-CCE and MAPPO on different scenarios within the GFR environment. Results for MAPPO are taken from Yu et al. (2021) Average and standard deviation success rates are reported over six random seeds for each scenario. S-MCTS results are based on our own implementation.

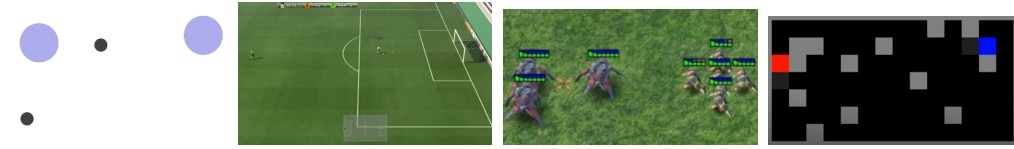

Figure 5: Left: MPE example observation. Middle-Left: GFR example observation. Middle-Right: Starcraft MA challenge example observation. Right: Laser tag example observation

that equilibrium approximation methods could converge onto a solution in a reasonable amount of time. Two scenarios are used: Goofspiel-6 (6-card variant) and Laser Tag.

**Google Football Research** Kurach et al. (2020). A team based mixed cooperative competitive football simulation environment. For this study we use smaller scale environments rather than the full game. We train agents to play as both teams and do not used fixed algorithms in our training process. Six scenarios are used: 3v1, CA(easy), CA(hard), Corner, PS, and RPS. We will also study three cooperative version of these scenarios.

**Multi Agent Particle Environment** Lowe et al. (2017); Mordatch & Abbeel (2017). A multi-agent particle environment that has a mix of cooperative, competitive, and cooperative-competitive tasks. We focus on three competitive and cooperative-competitive scenarios: Adv, Tag, and Push.

**Starcraft Multi-agent Challenge** Samvelyan et al. (2019) A multi-agent version of the popular real-time strategy game Starcraft. In this variant, all pieces on a single team are controlled simultaneously at each time step. Three scenarios are used: 3s vs 3z, 3s vs 4z, 5m vs 6m.

**Summary:** We will evaluate on 17 distinct games across four different environments, chosen for their open-source availability and prior evaluation by at least three algorithms in previous research.

## 5.2 COMPARED ALGORITHMS AND EVALUATION METRICS

We divide the set of algorithms we compared against into three main groups. We chose each of the algorithms tested because it is or was a recent state of the art algorithm for a respective environment, or it is a generally popular algorithm that serves as a useful benchmark; in addition, all algorithms we compared against need to be accessible with with open-source code.

**Equilibrium Approximation Algorithms**: NN-CCE (ours) compared against PSRO, JPSRO, CFR. Each of the latter three algorithms are not meant to scale to larger environments. Their application is limited to smaller competitive tasks from the OpenSpiel environment. For a comparison metric, we compared a direct head-to-head win rate of our algorithm versus the opposing algorithm for each scenario in OpenSpiel. All scenarios from open-spiel were symmetric 2p0s. In a single trajectory we labeled recorded the total points accumulated by our agent and the contemporary agent. The winner was determined as the side that accumulated more points. All algorithms in the comparison were trained and assessed over 10 random seeds.

**Cooperative Algorithms**: NN-CCE (ours) compared against MAPPO, MADDPG. Each of these algorithms are able to be applied to purely cooperative tasks within GFR and SMAC. They are both popular algorithms with tested open-source implementations. In addition they both demonstrate superior performance against a wide arrange of other contemporary algorithms Yu et al. (2021); Lowe et al. (2017). For a comparison metric, we compared total accumulated score in testing scenarios of our algorithm to the competitor.

**Competitive Algorithms**: NN-CCE (ours) compared against MADDPG, Simultaneous Move MCTS. All three of these algorithms can be applied to larger scale competitive scenarios within MPE, GFR and SCMAC. MADDPG has a tested open-source implementation. We implemented Simultaneous Move MCTS locally, and its detailed are found in the appendix. For a comparison metric, we compared a direct head-to-head win rate of our algorithm compared to the opposing algorithm for each scenario in the three environments listed. All competitive scenarios from MPE, GFR, and SMAC were asymmetric and two team-based.

|  | NN-CCE | jPSRO | PSRO | CFR | R |
|---|---|---|---|---|---|
| NN-CCE | - | **62%** | **70%** | **91%** | **100%** |
| jPSRO | 38% | - | 63% | 55% | 85% |
| PSRO | 30% | 37% | - | 52% | 73% |
| CFR | 9% | 45% | 48% | - | 74% |
| R | 0% | 15% | 27 % | 26 % | - |

|  | NN-CCE | C Q-Learning | JPSRO |
|---|---|---|---|
| NN-CCE | - | **100%** | **74%** |
| C Q-Learning | 0 % | - | 7% |
| JPSRO | 26% | 93% | - |
| Random | 0% | 36% | 12% |

Table 3: Win Rate on "Goofspiel-6" Scenario (Left) and "Laser Tag" Scenario (Right) from Open-Spiel. (Left) C Q-Learning results are not reported because it failed to beat the random opponent over many repeated trials. (Right) PSRO and CFR results are not reported because they failed to converge to a solution in a reasonable amount of time

# 6 PERFORMANCE RESULTS AND DISCUSSION

**Comparing to equilibrium approximation algorithms**. First, we apply our algorithm to the relatively small scenarios of Goofspiel and Laser Tag from the Openspiel environment. These results are summarized in Table 3. As can be seen, NN-CCE has a higher win rate across all three tasks compared to other equilibrium approximation algorithms. All algorithms we compared against do not scale to larger environments. Therefore, we compared them in the smaller scaled scenarios of OpenSpiel.

**Comparisons in competitive scenarios**. In the next set of experiments, we assessed NN-CCE on tasks that involved controlling multiple pieces in competitive environments: MPE GFR, and SMAC. In our assessment we compared against a popular multi-agent algorithm Multi-Agent MADDPG. The results of these experiments are summarized in Tables 2. Across all three environments (MPE, GFR, and SMAC), our algorithm had a higher win rate compared to MADDPG and SM-MCTS.

**Comparisons in cooperative scenarios**. In the final set of experiments we compare our method against popular cooperative algorithms over GFR and SMAC environments. We can see in table 1 that that our method, NN-CCE,

|  |  | MADDPG | S-MCTS |
|---|---|---|---|
| MPE | Adv | 82% | 83 % |
|  | Tag | 85% | 90 % |
|  | Push | 81% | 87 % |
| GFR | MA-PS | 60% | 100 % |
|  | MA-3v1 | 63% | 99 % |
|  | MA-C | 60% | 100 % |
| SMAC | 3s,vs,3z | 61% | 100 % |
|  | 3s,vs,4z | 64% | 100 % |
|  | 5m,vs,6m | 60% | 100 % |

Table 2: Win rate on MPE Tasks, Adv - Simple Adversary, Tag - Simple Tag Environment, Push - Simple Push. Win rate on "Google Football Research" Tasks. PS - Pass and Shoot Scenario, 3v1 - Academy 3 v 1 with keeper Scenario, Counter - Counterattack Easy scenario. All scenario descriptions can be found in the Google Football Repository. The tag "MA" refers to the multi-agent variant of the scenario, where multiple learning agents control all pieces, one agent per piece. Three tasks were chosen from the SMAC environment, which represent the number of pieces controlled by two players.

demonstrates marginal to high success-rate improvement over MA-PPO across 6 different scenarios in GR. *It is important to note that MA-PPO trains against a fixed opponent in scenarios where an opponent is present, such as scenario 3v.1, where as our agent trains against an adaptive policy (itself) in such a case.* Figure 6 also visualize the performance as a function of trajectories learned by our algorithm, MADDPG, and MAPPO.

**Discussion:** the key contribution of our work is to develop a *single* algorithm which is competitive or superior across each of the 17 benchmarks games we have tested. Our algorithm outperforms on 15/17 benchmarks, and for the remaining two (the cooperative PS and RPS scenarios in the GFR environment) it is competitive (within 1.5% of best algorithm performance). However, in those two cases, the best algorithm, which is MA-PPO, is trained against a human-coded opponent and thus requires an injection of human-knowledge.

# 7 CONCLUSION AND FUTURE WORK

In this study our aim was to create an algorithm that could perform well in multi-agent scenarios and trained through self-play with no human-knowledge injection. To accomplish this we proposed

a novel method to use NN to estimate a CCE for any given task. We demonstrate that our algorithm obtains higher performance against competitor algorithms inspired by game theory and deep MARL across a variety of benchmarks.

We demonstrated an improvement against other equilibrium estimation algorithms (PSRO, CFR) for smaller tasks, and and improvement against the current state of the art (MADDPG) for cooperative competitive environments in larger tasks. In conjunction, we also demonstrate a higher empirical consistency factor to our algorithm compared to MADDPG; our algorithm is much more likely to show improvement from baseline across MPE and GFR tasks. Lastly, we demonstrate that our algorithm also shows improvement over contemporary multi-agent algorithm MAPPO in purely cooperative tasks.

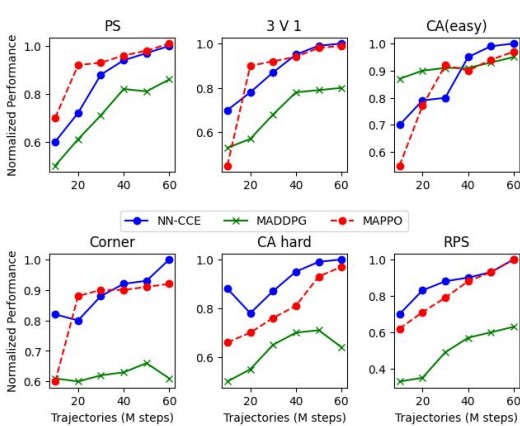

Figure 6: Results on GFR against a fixed opponent. NN-CCE (ours) and MADDPG are trained via self-play, MAPPO is trained against a fixed algorithm opponent.

The algorithm addresses two shortcoming of current MARL algorithms. Firstly, it can adapt to environments where agents with competing objectives exists. There is a small pool of algorithms that can successfully work in mixed cooperative competitive environments; of this pool our algorithm, NN-CCE, demonstrated higher performance across a variety of tasks. Secondly, it further further detaches itself from the need of human injected knowledge (typically used in the form of a human-designed agent to train against) and can therefore be used in environments where a strong fixed policy is not well known.

**Future work**. One clear drawback of our method is its limitation to discrete data tasks. Although we have higher performance than MADDPG in this realm, MADDPG boasts the ability to work in tasks with continuous space and continuous action spaces. A direct application of our method to continuous action spaces would not be advised, since it relies heavily on repeated visits to the same state (which will not naturally happen in continuous space).

Second, there is room to improve our method by creating value estimates that take into account more rewards from the environment itself. Currently, our value estimates are based only upon the rewards of the immediate next state and the estimate from the value network. However, we could obtain a much more accurate value estimate if we "unrolled" a trajectory and took into account the cumulative rewards of multiple time-steps into the future (a computationally expensive endeavor given the number of simulations our method requires). Unfortunately the solution is not as simple as one would hope. If the environment has sparse rewards, it could be that looking a few steps into the future would not yield any additional information (rewards), and thus we would have paid a computational cost without much gain. Also, there are experiments that must be conducted to ensure a proper integration of future rewards and the estimate from the value network so that the no-regret learning algorithm does not degenerate, which is something we have encountered in rudimentary implementations of this improvement.

**Reproducibility Statement and the Appendix**. We have made great efforts to ensure the reproducibility of our methodology section, and the appendix. While the methodology section only provides a brief overview of our method, the appendix describes in great detail each algorithm used in our methodology as well as the best parameters found through tuning and experimentation. In addition, the appendix contains a link to an anonymous code repository that contains an implementation of our method. There are instructions, found by following the link provided, that allow the download of code and training of agents using our method.

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

# A APPENDIX

## A.1 LINK TO PUBLICLY AVAILABLE IMPLEMENTATION

A link to a publicly available implementation of our work can be found here https://anonymous.4open.science/r/ImperfectInformationZeroSum-CA5C/CCE/Readme.md

## A.2 IN DEPTH METHODOLOGY

Our approach is based on two core ideas. The first idea is to train a separate neural network $\pi_i$ for each agent: this eliminates the curse of dimensionality. Indeed, although there are $|A|^n$ joint actions, by training $\pi_i : S \to \mathbf{A}_i$ for all $i = 1, \ldots, n$, we avoid an exponential growth of the action space. However, this approach comes with a tradeoff: we have effectively made agent decisions independent of each other – which is clearly highly sub-optimal. Indeed, agents could potentially gain from correlating their actions, which they cannot do under this strategy.

Out second idea is to mitigate the loss from this by using update rules which approximate a coarse correlated equilibrium (CCE); we view this as a "second best" solution to making correlated decisions across agents. Recall that a CCE is similar to a standard mixed Nash equilibrium, except that there is a *joint* distribution taken by all the agents which makes deviations gainless. By training agent policies to learn a CCE, we effectively bypass a major limitation of training independent policies. The idea is that even though the individual decisions are made separately, the agents are replying to each other, so that, in the limit, their time-averaged policies converge to a correlated action profile which is good in the sense of being a CCE.

This leads to the question of how to build dynamics that attain CCE in our setting. We build on recent work Daskalakis et al. (2022) which shows that EXP-IX, a standard algorithm in online learning with asymptotically vanishing regret, can learn CCEs. We thus replace the standard value estimation methods in MCTS based on UCB estimates with EXP-IV based estimates. Details are given below.

We call our algorithm NN-CCE. It is trained by iterating through three main steps: (1) gathering trajectories $D$, (2) processing trajectory information, (3) training a model on $D$.

We first gather $K$ trajectories in a given environment, each of length $H$. Next, we process our trajectories in reverse order. For the set of states in time step $H - 1$ **over all trajectories**, we train a Q-value network, $Q_{H-1} : s_{H-1} \times A \to \mathbb{R}$ to output Q-value estimates of state-action pairs. Using $Q_{H-1}$, we perform Multi-Agent EXP-IX to approximate a CCE policy and value estimate for every state over all trajectories at time step $H - 1$, $\{v(s_{H-1})\}_i$ and $\{p(s_{H-1})\}_i \forall i \in [K]$ (Algorithm 3).

After, we use $\{v(s_{H-1})\}_i \forall i \in [K]$, to train a new Q-value network $Q_{H-2} : s_{H-2} \times A \to v(s_{H-1}) \in \mathbb{R}$, and repeat CCE approximation and Q-value training until time step $h = 0$. In total, we would train $H$ Q-value networks per agent. Each Q-network takes as data the trajectory info for its respective time-step.

Finally, we train a policy model on all CCE policies calculated during our trajectory processing phase to form a final model that outputs a policy distribution.

A detailed algorithm is provided in Algorithms 1, 2, and 3.

## A.3 EXPERIMENTAL PARAMETERS

All Q-value and policy network parameters are given below. Let us define $I$ as the size of the input, $J$ as the size of the joint action space, $P$ as the size of the policy space, $H$ as the finite horizon, and $N$ as the number of players. All networks are trained using an Adam optimizer with learning rate $5e - 5$.

---

**Algorithm 1** NN-CCE approximation. This is a standard on-policy value-based method except we train a value network for each time step in reversed time step order.

---

**Input**: $G$, Stochastic Game
**Input**: $H$, finite horizon
**Input**: $K$, number of trajectories
**Input**: $M_H$, Q-value network
**Input**: $\pi_0$, Initial policy network

1: $R \leftarrow \{\emptyset\}$
2: **for** $t = 1, 2, \ldots, T$ **do**
3:      **for** $i \in K$ **do**
4:          $D_i \leftarrow \text{GenerateDataset}(\pi_i, G, K)$
5:          Let $b_{h'} \in D_i$ represent the set of all states from time-step $h'$ for player $i$
6:      **end for**
7:      **for** $h = H - 1, H - 2, \ldots, 0$ **do**
8:          $v_h, \pi_h \leftarrow \text{MA-EXP-IX}(s, M_{h+1}) \ \forall s \in b_h$
9:          $M_h \leftarrow \text{TrainValue}(\{v_{h'} : h' = h\})$
10:     **end for**
11:     $\pi_t \leftarrow \text{TrainPolicy}(\{\pi_h\})$
12: **end for**
13: return $R$

---

**Algorithm 2** GenerateDataset

---

**Input**: $G$, Stochastic Game
**Input**: $H$, finite horizon
**Input**: $K$, number of trajectories
**Input**: $\pi$, policy network

1: $\{b_h \leftarrow \{\}\}_{h \in H}$
2: **for** $k = 1, 2, \ldots, K$ **do**
3:      $s_1 \leftarrow \Delta(S)$                                $\triangleright$ sample starting state
4:      **for** $h = 1, 2, \ldots, H$ **do**
5:          $b_h \leftarrow b_h \cup s_h$
6:          $a_h \leftarrow \pi(s_h)$
7:          $s_h \leftarrow T(s_h, a)$                 $\triangleright T$ represents the transition function
8:      **end for**
9: **end for**
10: return $\{b_h \leftarrow \{\}\}_{h \in H}$

---

| Q-value Network Parameters | | | | |
|---|---|---|---|---|
| Sub-Network Name | Architecture | Learning Rate | L2-Regularization | Dropout |
| Representation Network | $[I, 256, 256, 32]$ | 5e-5 | 1e-4 | 0.5 |
| Q-value prediction Network | $[32 + J, 256, 256, S]$ | 5e-5 | 1e-4 | 0.5 |

| Policy Network Parameters | | | | |
|---|---|---|---|---|
| Sub-Network Name | Architecture | Learning Rate | L2-Regularization | Dropout |
| Representation Network | $[I, 1028, 1028, 64]$ | 5e-5 | 2e-4 | 0.6 |
| Q-value prediction Network | $[64, 1028, 1028, P]$ | 5e-5 | 2e-4 | 0.6 |

---

**Algorithm 3** MA-EXP-IX. This is a standard EXP-IX algorithm from Neu (2015) except we provide additional details because multiple players are all simultaneously using EXP-IX no-regret learning.

---

**Input**: $A$, Number of actions
**Input**: $T$, max time step
**Input**: $N$, number of players
**Input**: $Q$, Q-value estimation network
**Input**: $h$, current time horizon
**Input**: $H$, max time horizon
**Input**: $s$, state
**Output**: weight matrix, $w$ and value estimate, $v$

1:   $w \leftarrow \overrightarrow{1} \in \mathbb{R}^{N \times A}$
2:   $v \leftarrow \overrightarrow{0} \in \mathbb{R}^{N}$
3:   **for** $t = 1, 2, \ldots, T$ **do**
4:      $j \leftarrow \{\}$
5:      **for** $n = 1, 2, \ldots, N$ **do**
6:         $p_{t,i,n} = \frac{w_{t,i,n}}{\sum_{j=1}^{A} w_{t,j,n}}$
7:         Draw $I_{t,n} \sim p_{t,n} = (p_{t,1,n}, p_{t,2,n}, \ldots, p_{t,A,n})$
8:         $j \leftarrow j \cup \{I_{t,n}\}$
9:      **end for**
10:    **if** $h = H$ **then**
11:      Observe loss $l_{t,j} = (l_{t,I_{t,1}}, \ldots, l_{t,I_{t,n}})$
12:    **else**
13:      $l_{t,j} \leftarrow Q(s, j)$
14:    **end if**
15:    **for** $n = 1, 2, \ldots, N$ **do**
16:      $v_n \leftarrow v_n + l_{t,t,n}$
17:      $\widetilde{l_{t,i,n}} \leftarrow \frac{l_{t,i,n}}{p_{t,i,n} + \gamma} \mathbb{I}_{\{I_t = i\}}$ for all $i \in [A]$
18:      $w_{t+1,i,n} \leftarrow w_{t,i,n} e^{-\eta \widetilde{l_{t,i,n}}}$ for all $i \in [A]$
19:    **end for**
20: **end for**
21: return $w$ and $v$

---

For a given environment, we trained a total of $H * N$ Q-value networks, one for each player and time step. The number of Q-value networks could be reduced if the game was fully competitive or fully cooperative. In this type of environment, only $H$ Q-value networks were trained. For every environment, a total of $N$ policy networks were trained.

### A.4   Factors influential to NN-CCE Performance

In this section we provide a series of ablation studies to demonstrate the factors that affect the performance of our NN-CCE algorithm. Results are given comparing different versions of our agent against a fully random algorithm and MADDPG in small test scenarios within the multi particle environment. The specific scenario is "simple" with 2 adversarial agents and 1 good agent. Results are given in terms of the total reward accumulated during a testing episode. Where either trained MADDPG or CCE agents control the adversarial players, and the other controls the good player.

These results guided our development and implementation, contributing to the overall success for the larger tasks and test beds.

#### A.4.1   Number of trajectories

Our agent uses significantly less environment interactions compared to the MADDPG algorithm. Because we do significant processing for each trajectory before neural network training, we restricted the number of trajectories sampled to at most half of that of MADDPG.

We then trained MADDPG on a three small test scenarios in MPE and recorded its performance after 10k, 30k, and 50k trajectories through the environment, one for each test scenario. Our algorithm

was then trained on the same small test scenarios but limiting the trajectories to 5k, 15k, and 25k trajectories respectively.

For each small scenario, our algorithm scored at least $10\%$ better than MADDPG using at most half of the number of trajectories through the environment. For these scenarios we had NN-CCE play against MADDPG agents in a head-to-head however we recorded the average score

In addition, the ratio of failure cases for MADDPG grew with the number of trajectories. We define a failure case as an instance where the policy post training does not improve significantly beyond the performance of a random policy.

Interestingly, the ratio of failure cases for our agent decreased significantly as we increased the number of trajectories. Our agent makes a trade-off compared to other RL algorithms: we sample less trajectories from the environment but spend much longer processing the trajectories we do sample using no-regret learning.

### A.4.2  DIVERSITY OF NODES WITHIN A LAYER

Given the results from the previous section, we take significantly less trajectories through any environment we are training NN-CCE approximation in. As a result, we initially observed a wider range of performance for NN-CCE on the same environment over many random seeds.

We discovered that NN-CCE agents that performed higher tended to have a higher spread in the value estimates for nodes across every layer. We measure spread for value's in each layer using the coefficient of variation (CV) for a given layer: $\frac{\sigma_h}{\mu_h}$.

In order to utilize this observation as a reproducible process, we developed a subroutine within training that generates a fixed number of trees, measures the CV for each tree and a given layer, and uses the tree with highest CV for that layer.

While this subroutine does increase the total number of environment trajectories, the agent still only trains on one of the trees generated. In addition this subroutine did not increase the maximum score in any of the test environments, instead it made the scores more consistent (less failure cases).

### A.4.3  STRATEGIC DOMINANCE ACTION PRUNING

The goal of no-regret learning is to grow the regret with respect to the best action in hindsight sublinearly. The algorithm will converge on to what it evaluates as the best action. If all players utilize no-regret learning, then their learned policies converge to the set of CCEs.

Objectively speaking at a given state with N-players each having K strategies, there are multiple CCEs that exist, and our agent would converge to one of them. In fact we also found that successful MADDPG algorithms would converge onto one CCE for a given state as well.

In classical game theory, having two or more competing equilibrium's for a state is not a problem as if equilibrium A was strictly better than equilibrium B, equilibrium B would not be an equilibrium by definition, but in MARL it can be an issue.

This is because the very definition of equilibrium assumes that all other players follow that recommended equilibrium. But in some cases of MARL, such as when our agent controls the 1 good agent, and MADDPG controls the 2 adversarial agents. Suddenly that assumption is violated, and the performance of our agent is due to random chance on how well our equilibrium compares to the opponent equilibrium; in scenarios like this, more than one player is not following the equilibrium we learned.

Therefore it is important not only to learn how to play one equilibrium, but also learn to adapt to different equilibrium's for a given state.

One way we found to improve performance around this problem is to prune the dominated strategies for all players before no-regret learning. Strategies that were deemed to be dominated by any other strategy were masked and not allowed for selection and their weights were ignored when converting weights to policy, thereby receiving a probability of selection of 0.

We found that this optimization did not increase the maximum performance of our agent against MADDPG across any test scenario, but it did increase the mean performance against MADDPG by 23% from an average score of 15.3 to 18.8 over many repeated test episodes and 10 random seeds.

### A.4.4 JOINT VS INDIVIDUAL POLICY OPTIMIZATION

Joint policy optimization has poor scaling to larger tasks but it allows for much higher express-ability of policies especially for competitive tasks, or tasks with competitive elements compared to optimizing solo policies. The problem is that in team based competitive tasks, if we view the player policies is random variables, the players should not be viewed as independent variables. Instead, they should be viewed as dependent variables (all players on the same team). A joint distribution created by multiplying the player marginal distributions cannot come close to the complexity of a joint distribution over all possible actions. It does not allow for coordination amongst the players and never will allow for such coordination. This is a problem for exploitability of a strategy in competitive settings.

In cooperative tasks, this lack of express-ability isn't actually that much of a problem. Because we are just looking for the best joint action as a needle in the haystack, we don't need to consider All joint action futures individually, but can consider each players policy and form a joint distribution by multiplying the the marginals together (treating the players as independent).

### A.4.5 IMBALANCED DATA SETS FOR EQUILIBRIUM VALUE PREDICTION

One of the highest impacting aspects of predicting unknown states' equilibrium values is the imbalance in value estimates accrued during simulation. This is particularly true in the first iteration where the policy sampling the states is effectively a random policy.

We can see here for the MPE environment allowing a random policy to sample leads to the following value distributions. It is heavily skewed in favor of the value 0, as it is very unlikely for random policies to collide leading to non-0 rewards for any given trajectory.

When we attempted to train a Q-value network on this heavily biased data set, we found that, as expected, it primarily predicted a value of 0 for testing data; the errors for non-zero testing data was extremely high and variable.

Therefore we applied up-sampling to minority values. Initially we separate the continuous value training data, (X,y) into K classes. Each class is defined as a non-overlapping range of size range(y)/K. In order to prevent over representation of a small set of data points during sampling, we maintain a rule that each class except one, $k_s$ must contain at least 1000 data points, where $k_s$ is defined as the class with the least data points. We therefore recursively combine the two smallest classes until the condition is met.

### A.4.6 TRAJECTORIES

There is one novel crucial component within the game tree creation algorithm: the partially random off-policy trajectories. In this component, a subset of players follow their policy but the rest are randomized. We found that a mix of on-policy and randomized learning agents improved the average performance of the final agent, compared to fully on-policy and fully random training, across multiple tasks within the MPE environment.

For the purpose of score comparison between the three variations, we normalize the average and standard error of the fully random training performance to $1 \pm 0.17$, respectively. Fully on-policy training yielded an average performance of $0.8 \pm 0.3$, while partial randomization yielded an average performance of $1.3 \pm 0.2$.

We attempted to use common place exploration vs. exploitation methods, such as UCB score from MCTS and epsilon learning from DQNs to improve performance, but neither significantly impacted performance in our simultaneous-move multi-agent setting.

### A.4.7 NN SUPPORTED POLICY AND VALUE ESTIMATION

At the core of our algorithm is the estimation of non-stationary policies and values for any state and time pairing. For any given state, $s$, at time step $h$, we run a neural network supported bandit algorithm for a set number of iterations. During the bandit algorithm, we accumulate an average value for $s$, and after the bandit algorithm we obtain our policy for $s$. Once all states within time step $h$ have been processed, we train a NN, $N_h$, on the states of time-step $h$, and repeat the process for nodes in time-step $h - 1$ using $N_h$ as our supplementary network.

Compared to other MARL algorithms, such as MADDPG, PPO, or deep-MCTS, we trade-off a higher volume of data for attempting to get higher quality data. We can directly compare the effects of this layer by layer approach by comparing our results to those of stationary policy estimation on multiple tasks within the MPE environment.

In stationary policy estimation, we no longer take a layer by layer approach, but instead accumulate value estimation by back propagating visited leaf nodes in the tree through the reverse trajectory used to reach them. Policy estimations, rather than using bandit algorithms at each state, are accumulated by visit count to each successor state. This approach is very similar to the approach in deep-MCTS.

By normalizing the average and standard error of the stationary estimation performance to $1 \pm 0.56$, we find that our layer by layer method yields more robust and higher quality results with an average performance of $2 \pm 0.23$.

### A.4.8 DATA PROCESSING AND LEARNING OBJECTIVES

After generating a game tree (Algorithm 4) we have a large data structure of states seen through environment interaction organized by time horizon. We will refer to "layer h" of our tree as as all nodes that are $h$ time steps away from any of the root nodes.

Contrast to monte carlo tree search based algorithms, we do not attempt to approximate a value and policy for each state during the search itself, instead we process all nodes in reversed time order once the tree has been fully created.

SM multi-agent reinforcement learning opens the question as to how we should estimate both the value and policy of a given state. In classic RL and perfect information multi-agent RL, the value of a state is estimated using the bellman equation:

$$V_\pi(s) = R(s) + \gamma * max_a V_\pi(s')$$

where $s' = T(s, a)$, and the policy can be determined by picking the action that leads to the next state with highest value for every state.

The application of bellman equation to SM MARL becomes translucent as we would need to know the policy of all agents in order to make a value estimation. As mentioned previously, other MARL algorithms address this issue by using a non-learning policy for all other agents (MA-PPO), or keeping a local policy estimation for all other agents (MA-DDPG). Both of these options lead to potentially sub-optimal generalization as the performance of the agent is then directly tied to the opponent they trained with or against.

In our study we begin by processing the nodes in layer $h = H$, the terminal nodes of the tree. Nodes in layer $H$ have the unique property that $V(s \in S_H) = R(s \in S_H)$ regardless of any policy, since the episode terminates after reaching state $S_H$.

We begin by updating the associated weights for all nodes in layer $H$ where the loss of a given state, $s$, for player $n$ $l(s, n) = \frac{r_n(s)}{\max_{s \in S_H} r_n(s)}$ using Algorithm 3, which returns a set of policy weights and state value estimates, $W_H, V_H$.

As we are at $h = H$, we are only concerned with the value estimates, $V_H$. Combining these value estimates with the states of their parent nodes, $S_{H-1}$, we create a dataset for a Q-value estimation network of tuples $(s, a, s', v)$ where $s \in S_{H-1}$, $s' \in S_H$, $a$ is a valid action such that $T(s, a) \to s'$, and $v := V(s') \in V_H$, and train a network on this data set. The process is then repeated until we reach $h = 1$.

---

**Algorithm 4** Generate Game Tree

---

**Input**: $S$, State Space
**Input**: $\Delta$, Starting State Distribution
**Input**: $S_0 \subseteq S$, Set of starting states
**Input**: $A$, Action Space
**Input**: $H$, finite horizon
**Input**: $K$, number of simulations
**Input**: $T$, Transition Function
**Input**: $M$, Neural network model
**Output**: $R$, Set of root nodes

1: $R \leftarrow \{\emptyset\}$
2: **for** $k = K, K - 1, \ldots, 0$ **do**
3:      $s_0 \leftarrow \Delta(S_0)$
4:      **if** $s_0 \notin \{s' : N.state \in R\}$ **then**
5:          $v_0, p_0 \leftarrow \text{Predict}(M, s_0)$
6:          $N \leftarrow Node(s_0, v_0, w \leftarrow p_0)$
7:          $R \leftarrow R \cup \{n\}$
8:      **end if**
9:      $N \leftarrow \text{GetNode}(R, s)$
10:     $s \leftarrow s_0, w \leftarrow p_0, \tau \leftarrow 0,$
11:     $N.n \leftarrow N.n + 1$
12:     $children \leftarrow \text{GetChildren}(N)$
13:     **while** $\tau = 0$ **do**
14:        Sample joint action, $j$, using $\frac{w_i}{\sum w_i}$ for all players
15:        $s' \leftarrow T(s, j)$
16:        **if** $s' \notin children$ **then**
17:           $v', p' \leftarrow \text{Predict}(\text{Model}, s')$
18:           $N' \leftarrow Node(s', v', w' \leftarrow p')$
19:           $children \leftarrow children \cup \{n'\}$
20:           $\tau = 1$
21:        **else**
22:           $s \leftarrow s', w \leftarrow w'$
23:           $children \leftarrow \text{GetChildren}(N')$
24:        **end if**
25:     **end while**
26: **end for**
27: return $R$

---

## A.5 SIMULTANEOUS-MOVE MCTS

We implemented our own form of Simultaneous-move MCTS based on the algorithm used in MuZero.

It was adapted to fit simultaneous-move tasks where both teams/players picked a move simultaneously before it is sent to the environment. Value was calculated as an accumulated value for all iterations at a node, and policies were generated by child visit count.

### A.5.1 DATA STORAGE AND SAMPLING

After the tree is generated we save every node to a replay buffer. Each node in the replay buffer is saved as a tuple defined as Transition(Node):

1. $\{O_p\}$ for $p = 1...n$

2. Node value

3. Node policy

4. Node time step

5. Node player

6. Node Visit Count

Where $O_p$ is the observation for each player at Node. Because the goal of the NN is to replicate the tree value and policy estimation, we sample such that every node in the same time step has a uniform chance of selection. In addition we give each time step equal chance of selection. For instance if there is a task with 5 time steps, the overall probability would be divided to 20% per time step. Within each time step, the 20% is divided evenly amongst the nodes associated with that time step. Note that it is likely different time steps will have a different number of nodes. A further discussion of this method is given in the ablation studies.

We also do not use priority sampling with our replay buffer. It is an extremely popular method that has been shown to drastically increase the speed of training (Schaul et al., 2015). However we believe it does not fit well with our data, and were not impressed with its early empirical results on training.

Our data is atypical to other reinforcement learning algorithm. Typical RL algorithms store trajectory tuples of the form $(s, a, s', r)$, however we store information that has been aggregated over many iterations. Therefore the problem of rare experiences, that prioritized buffers address, is not as applicable. Sampling uniformly across layers provides better value and policy estimates in future iterations.

### A.5.2 MACHINERY

Neural networks and scenario simulations were performed on a local machine containing a NVIDIA GeForce RTX 2080 Ti graphics card (GPU) with 11gb of memory. In addition the machine contains a 3.70GHz Intel(R) CPU with 8 cores.

### A.5.3 NEURAL NETWORK TRAINING

Neural network architecture follows a standard DQN structure. It takes as input a batch of local observations and outputs a value estimate as well as policy estimate. There is a representation network that is malleable depending on the input type. If the observations are images then it is generated as a convolutional NN, otherwise it is a deep fully connected NN. The number of layers in the representation network are flexible to the performance needs, but we find 8 layers each of size 256 to be sufficient for all tasks in this study.

After the data is passed through the representation network it is then passed through two separate networks, the value and policy network. The value network takes the representation output and produces a vector of variable size (support size) which is then converted to a scalar value. This method of estimating a value using a vector operation was popularized in Schrittwieser et al., 2019. It causes outputs to be between 0 and 1 and thus aids in numeric stability. The policy network also takes as input the representation output and produces a vector the size of the max action space of all agents.

During training, the network predictions for each observation are measured against their stored node information counterparts stored in the replay buffer. We use cross entropy loss with stochastic gradient descent to optimize the network. Further implementation parameter details can be found in the appendix.

The total procedure involves iterating between tree generation and neural network training. In order to measure intermediary progress, we measure the performance of our network against a random agent after each training session. We use the score against the random agent as both a validation procedure and termination procedure. If the agent scores worse against the random agent on an iteration, the training is undone and the iteration repeats itself. This is common procedure for iterating algorithms such as deep MCTS. If the network does not significantly improve its score after a series of iterations, the training process is terminated, and the last updated model is output.

### A.5.4 SAMPLING TRAJECTORIES AND GENERATING A GAME TREE

We define a game tree is a series of Nodes, each representing a state, and directed edges, representing a transition between nodes. Algorithm 4 provides a detailed account of how the game tree is generated and stored.

The algorithm returns a set of nodes, $R$, which contains the root node of $|R|$ trees generated in simulation. There is a unique tree for each unique starting state. If a task only has one unique starting state, then $|R| = 1$.

Each node, $N$, stores the following information: $s$, state, $v$, value, $w$, policy weights, $children$, set of edges to child nodes, $parent$, edge to parent node

The simulation begins by choosing a root node from the starting distribution (line 3). If the state is new, we create a new root object and store it in our set of roots, $R$ (lines 4 - 7). If the state has been seen in a previous simulation, then we get the corresponding node object (line 9). In either case, we get the set of children and other variables (lines 10-12).

The next step is to reach a leaf node. We define a leaf node as either a terminal node, or a node that does not yet exist in the tree. Upon reaching a leaf node, we create a new node and set the appropriate attributes (lines 16-19).

