# OpenReview forum: "Tree Search for Simultaneous Move Games via Equilibrium Approximation"
_ICLR.cc/2025/Conference — ICLR 2025 Conference Withdrawn Submission_

### Official Review · Reviewer_p4jF · 2024-11-02

**Soundness:** 2
**Presentation:** 2
**Contribution:** 2
**Rating:** 3
**Confidence:** 5

**Summary:**

***Summary of paper** This paper considers deriving tree-search algorithms in perfect information simulatenous-move games. They derive one search algorithm by approximating coarse correlated equilibrium using an online learning formulation.

**Strengths:**

The presentation is clear.

**Weaknesses:**

The technical contribution is not sound.

**Questions:**

First, there have already been works that consider monte-carlo tree search in simultaneous-move game, where this paper did not cite and discuss:

[1] Monte Carlo Tree Search in Simultaneous Move Games with Applications to Goofspiel, Lanctot et. al.

[2] Monte Carlo Tree Search Variants for Simultaneous Move Games, Tak et. al.

[3] Convergence of Monte Carlo Tree Search in Simultaneous Move Games, Lisy et. al.

In these works they devise MCTS-style algorithms using UCT for explorations and achieved relative good results. However in the current paper the authors did not apply UCT. Could the authors also discuss your methods to the above. Also I don't see any of the above methods being compared in the evaluation section.

Second, in recent works about Diplomacy, they have already derived search-method based on approximating CCE-like objective [4, 5]

[4] No-press Diplomacy from Stratch. Bakhtin et. al.

[5] Mastering the Game of No-Press Diplomacy via Human-Regularized Reinforcement Learning and Planning. Bakhtin et. al.

In the above works they use regret-matching for computing approximate equilibira and use Nash-Q learning for iteratively learning policy and value functions. Could the authors compare your approache with theirs.

---

### Official Review · Reviewer_7Bms · 2024-11-02

**Soundness:** 2
**Presentation:** 1
**Contribution:** 2
**Rating:** 5
**Confidence:** 4

**Summary:**

This paper explores the adaptation of neural network-supported tree search, commonly used in perfect information games, to simultaneous-move games, a subset of partial information games. In these games, both agents know the game state except for the opponent’s immediate move, revealed only after each agent's move, as seen in benchmarks like Google Research Football and Starcraft. The authors propose a novel method to approximate a coarse correlated equilibrium within tree search, allowing the algorithm to perform effectively across cooperative, competitive, and mixed tasks. Results show that the approach outperforms leading MARL algorithms across widely accepted benchmarks.

**Strengths:**

1. The proposed depth-limited scheme is well-suited for simultaneous-move games.
2. Extensive experiments validate the method across various scenarios.
3. The method allows for better parallelization, making it more practical for real-world games.

**Weaknesses:**

1. The paper's writing and formatting are poor, with some images, tables, and equations arranged in a cluttered, two-column layout (e.g., Figures 1 and 4, Tables 1 and 2, and Equations 1-5), making the document look disorganized. Additionally, citation formatting is incorrect; in the ICLR template, \citep should be used instead of \citet when authors or publications are not part of the sentence.
2. The paper lacks theoretical support. While it spends much space arguing that CCE is superior to min-max, it does not demonstrate that the combination of EXP3-IX and depth-limited d-MCTS can ultimately converge to CCE.
3. The terminology is somewhat unprofessional. Typically, simultaneous-move games are not categorized as partially observable games, as standard Markov or stochastic game definitions allow for simultaneous moves. If the authors intend to contrast with traditional perfect information games, they should instead use “imperfect information games”.
4. The paper makes several unsupported claims. For example, in line 35, the assertion that "successes in partially observable settings have been more muted" overlooks numerous well-known works in this setting (e.g., DeepStack, AlphaStar, OpenAI Five, DeepNash). Similarly, line 46 claims that "playing according to a CCE gives you performance guarantees against any opponent in competitive tasks," which is inaccurate—CCE does not ensure performance guarantees except in 2P0S games, where it aligns with Nash equilibrium.

**Questions:**

1. Given that NN-CCE limits sampling to child nodes at a depth of only one layer, could an alternative approach be to perform weighted summation directly based on action probabilities? This might avoid redundant sampling steps.
2. Since the method employs Monte Carlo sampling, would it be suitable to compare it with similar techniques, such as Monte Carlo CFR?

---

### Official Review · Reviewer_iwW1 · 2024-11-04

**Soundness:** 2
**Presentation:** 2
**Contribution:** 2
**Rating:** 3
**Confidence:** 4

**Summary:**

The paper introduces a method for approximating coarse correlated equilibria (CCE) in multi-agent reinforcement learning (MARL) settings with perfect information and simultaneous moves (Stochastic games aka Markov games).

**Strengths:**

There was a lot of effort put into designing the algorithm, running many experiments, and writing the paper.

**Weaknesses:**

The paper has several weaknesses which prevent me from recommending it for acceptance.

I think the actual description of the algorithm is unclear. At inference time, what is the tree-search algorithm? Appendix A.5 suggests that there will be a game tree and an MCTS-like algorithm (as does the title of the paper), but Section 4 suggests that the method is no-regret algorithms using 0-step lookahead (just q-values)?

The experiments performed also don't test the core hypothesis of the paper -- that the algorithm computes a CCE. The experimental results are also fishy, and hyperparameters are not given for the baselines. Perhaps my biggest issue with the paper is that in the experiments, CFR does not perform well on "Goofspiel-6", but one would expect CFR to be able to compute a close Nash equilibrium in the game. The experimental code for CFR does not seem to be in the code linked in the appendix.

The paper's driving motivation is convergence to CCE. However, the experiments measure head-to-head winrates. Head-to-head winrates already seem less preferable than head-to-head EV (since agent 1 may have +EV against agent 2 despite having a lower winrate). However, to measure convergence to CCE in small, 2P0S games, one could simply compute exploitability.

---

The paper should cite other methods that use neural nets and tree search to compute a CCE.
- For example, ReBeL [0] and Deepstack/Student of Games [1] use CFR, where the players' marginal time-averaged strategies form a N.E. in 2P0S games, because their joint time-averaged strategy is a CCE [2].
- Research in Diplomacy, a simultaneous-move multiplayer game, also uses neural nets, lookahead, and no-regret dynamics: [3], [4].
- There is also a rich line of research into q-learning for equilibria in stochastic games. Although mostly in the 2-player setting, the approach is similar to this paper and should be cited. The research starts with Littman's seminal minimax q-learning, and the related works section of this paper ([5]) gives a good overview of modern developments.

I believe this existing research contradicts the statement in the abstract: "Neural network supported tree-search has shown strong results in a variety of perfect information multi-agent tasks. However, the performance of these methods on partial information games has generally been below competing approaches."

---

> This approach works well for zero-sum, perfect information games like chess or Go, but when we move into the realm of partial information games, the min-max paradigm becomes inappropriate.

I would say the min-max paradigm is still valid in partial information games. The difference is between 2P0S and general-sum/multiplayer. Also, the paper should mention somewhere that the "productized" policies (each player playing their own marginal, time-averaged policies) of a CCE are Nash equilibria in the 2P0S setting.

---

> The limitation for equilibrium approximation algorithms is that they are not easily applied to tasks with larger state and action spaces. The majority of testing regarding such algorithms have been on small tasks.

I don't think this is true.

---

> Using the regret bound provided by Neu (2015), it becomes clear that 800 time steps is insufficient for this method. At 800 time steps, the theoretically guarantees provided by EXP3-IX are very poor: we have not yet found the best action.

Why do we refer to a "best action"? Why do we expect that action probabilities stabilize? The guarantee is convergence to a CCE, not a "best action" (which isn't well-defined), no? A CCE is a *distribution* over *joint* actions.


[0]: https://arxiv.org/abs/2007.13544
[1]: https://www.science.org/doi/10.1126/sciadv.adg3256
[2]: https://arxiv.org/abs/2310.11518
[3]: Learning to Play No-Press Diplomacy with Best Response Policy Iteration
[4]: https://arxiv.org/abs/2210.05492
[5]: https://arxiv.org/abs/2306.05700

**Questions:**

See above.

---

### Official Review · Reviewer_PxSJ · 2024-11-08

**Soundness:** 3
**Presentation:** 3
**Contribution:** 3
**Rating:** 8
**Confidence:** 3

**Summary:**

This paper develops a method that combines deep Monte Carlo Tree Search with online no-regret learning in order to approximate coarse correlated equilibria in both cooperative and competitive simultaneous move games.

**Strengths:**

- The related work is strong, and the paper is reasonably well motivated
- The outline of the methodology is clear
- The motivation for using the simulation depth of 1 is strong
- The results are overall very strong, however I would have liked to see some more of the PSRO variants that focus on other aspects (e.g. population diversity) that may be stronger performing baselines than the more standard PSRO and jPSRO, especially in larger scale environments. However, the general strong performance over both competitive and cooperative tasks is impressive.
- Overall I like the general simplicity of the approach in terms of the high-level methodology. Furthermore, I think the empirical results are strong on a fairly standard set of baselines.

**Weaknesses:**

- It is a little difficult to follow the first part of section 4.1
	- e.g. the writing suggests the value network only takes joint actions as inputs, I assume it also takes the state? The equations 1 through 5 could also use a bit more explanation.
- I am not sure about the argument made that the PSRO methods are not designed to work in large environments - e.g. Towards Unifying Behavioural and Response Diversity for Open-ended Learning in Zero-Sum games (Liu et al. 2021) applied a diversity aware PSRO to Google Research Football and subsequent PSRO papers have done the same
- My main concern with the paper is that whilst the body of the paper provides a good high-level overview of the framework, some potentially key details for both understanding (e.g. The explanation of EXP3-IX usage is limited and it is difficult to follow how one would implement it) and re-implementation are buried in the appendix / the provided code.

**Questions:**

- Line 104 - what is a SM game?

---

### Author Response · Authors · 2024-11-28
**Thank you to the reviewers**

We have decided to withdraw this submission.

We first wanted to thank each reviewer for their comments and feedback regarding our submission; they will prove extremely useful as we look to improve the quality of our work and its presentation.

---

### Note · Authors · 2024-11-28

I have read and agree with the venue's withdrawal policy on behalf of myself and my co-authors.